# Effect of foliar application of seaweed (organic fertilizer) on yield, essential oil and chemical composition of coriander

**Ayse Ozlem Tursun** *

University of Malatya Turgut Ozal, Battalgazi Vocational High School, Battalgazi, Malatya, Turkey

* ayoztursun@hotmail.com

## Abstract

The effect of fertilization on the yield of medicinal and aromatic plants is important. Among various fertilizers, seaweed is an environment friendly organic fertilizer. This study was conducted to determine the effects of foliar application of different doses of seaweed extract on the yield and essential oil content of coriander. Application was done before and at the beginning of flowering at two different locations. Results showed that the essential oil content was higher in the location with high sand content while the yield and yield components were higher at the location with low sand content and high organic matter content. Seaweed application at a dose of 2 ml L$^{-1}$ showed positive effects on the growth, development, yield and essential oil contents of coriander plant. Linalool (%) was determined as the most important essential oil compound and the foliar application of seaweed showed positive effects on the linalool percentage. It can be stated that the foliar application of seaweed (organic fertilizers) has beneficial impacts in terms of increasing the yield and quality traits of coriander.

## Introduction

Organic fertilizers have an important role in improving the soil quality by providing basic macro and micronutrients [1]. Organic fertilizers can be applied as foliar spray or directly to soil for improving crop growth and quality.

The limited availability of soil fertilizers makes the foliar fertilizers more important [2]. In recent years, the importance of organic fertilizers or biostimulants has increased due to several reasons. Bio-organic matters or biostimulants that are known as biofertilizers are used for improving nutrient intake, stimulating growth and rising plant tolerance to environmental stresses [3,4]. The seaweed extracts that are considered as biostimulants have positive effects on the growth and development of plants [5]. In addition to improving plant growth, seaweeds are also used for increasing yield and quality, and improving chemical composition of secondary metabolites [6]. Being environment friendly, natural organic products such as seaweed fertilizers are gaining popularity for a wider use in agriculture [7]. Liquid extracts obtained from seaweeds have been recently used as foliar fertilizers for many plants [8]. With the use of seaweed extracts for agricultural purposes, it was claimed that these extracts had positive effects on germination, root growth, leaf size, tolerating unfavorable soil conditions and, nutrient

**Data Availability Statement:** All relevant data are within the paper.

**Funding:** The author(s) received no specific funding for this work.

**Competing interests:** The authors have declared that no competing interests exist.

uptake from the soil [9,10]. When the seaweed foliar fertilizers are applied by spraying on the plant leaves, they can increase cell division and growth rate and, these fertilizers can also be applied as liquid on the soil surface. Seaweeds are rich resource of micro and macronutrients, amino acids, vitamins, which can affect positively cellular metabolism, plant growth, and yield [7]. The most commonly used seaweed as a biostimulant is an alga, obtained from North Atlantic Ocean and known as *Ascophyllum nodosum*, rich in polysaccharides (alginate, fucoidan and laminarin), minerals, and vitamins. Besides, they are rich in bioactive compounds like polyphenols, lipids, and proteins [11]. Chemical analysis of seaweeds and their extracts demonstrates the presence of diversified plant growth regulators likes auxins and cytokines in various amounts [12]. Furthermore, it is considered that extracts of seaweeds might be used as a replacement of chemical fertilizers [5].

Coriander (*Coriandrum sativum* L.) belongs to the Apiaceae family, which is one of the oldest medicinal plants that have been cultivated since centuries [13,14]. The coriander plant has been originated from Mediterranean region, mostly cultivated in South Asian countries, and also cultivated in high mountain Middle East Europe and Southeast African regions [15,16]. The most used part of coriander plants are their fruits that have essential oils and used in cosmetic, medicine, and spice industries, and its fresh and dry leaves are used as spice and vegetables, also its grounded mature fruits are directly used as spice [17]. The essential oils obtained from the fruits are used in food, alcohol, and cosmetic industries. Additionally, the essential oils of coriander have antibacterial, antioxidant, antidiabetic, antidepressant, antifungal, antihypertensive, anticancer, improving memory, antimutagenic and diuretic effects [18–20]. There are 0.03–2.7% essential oils in matured coriander fruits. The most important component of essential oil is linalool, which is almost 50 to70% of the whole essential oil compounds in a coriander plant [21–24]. Due to its valuable ingredient of linalool, coriander essential oil is used in the cosmetic industry. Genetic structure, climate conditions and agricultural practices can cause some changes in ingredients and components of essential oil [22,23,25].

The importance of chemical and organic fertilizers for getting high yields from medicinal and aromatic plants has been reported in several studies [26–28]. The demand for the organic products has been continuously increasing all over the world including Turkey. Because of this reason, use of organic and microbial fertilizers is being considered seriously all around the world. However, there are very few studies about the effects of organic fertilizers on medicinal and aromatic plants. Hence, this research determined the effects of application time and different doses of seaweed fertilizer on yield, quality and essential oil components of two coriander varieties. Moreover, it was also studied that how seaweed doses change the linalool content and concentration in coriander.

## Materials and methods

### Site description

Field experiments were conducted at Malatya Turgut Ozal University Malatya, Turkey (39.01.51˚N, 38.29.15˚E) during the growing season of 2021. Coriander was grown at two different locations (location A and location B) which had different soil structure. The soil properties have been provided in the Table 1. The soil analysis showed that at location A, the soil type was sandy-clay loam and it was rich in organic nutrients but the soil type of location Bwas sandy and poor in organic nutrients (Table 1).

Weather data during the growing seasons and the long-term weather were obtained from Turkish General Directorate of Meteorology (Table 2). Annual average temperature during the growing season for 2021 and the long-term temperature were 25.4˚C and 23.5˚C, respectively while annual precipitation were 58.8 mm and 78.5 mm, respectively (Table 2).

**Table 1. Physical and chemical properties of the soils in the experimental areas (locations) (0–20 cm).**

| Compounds | Locations | |
|---|---|---|
| | Location A | Location B |
| Sand (%) | 38 | 64 |
| Silt (%) | 22 | 10 |
| Clay (%) | 40 | 26 |
| pH | 8.6 | 8.7 |
| EC (dS m$^{-1}$) | 0.3 | 0.2 |
| Organic matter (%) | 2.6 | 1.2 |
| CaCO$_3$ (%) | 36.1 | 28.3 |
| P$_2$O$_5$ (kg ha$^{-1}$) | 9.7 | 11.7 |
| K$_2$O (kg ha$^{-1}$) | 65.1 | 63.8 |

## Experimental design

Soil preparation was done according to local practices for coriander production. In both the experimental fields, soil was cultivated using a chisel plough, followed by a disk harrow and finally by a harrow to obtain a smooth seedbed before sowing the crop. Erbaa and Gamze coriander varieties were used in the experiments and both the varieties were sown on 29 May 2021, with a 25 cm space between rows and 5 cm between plants. At the time of sowing, 50 kg ha$^{-1}$ N and 50 kg ha$^{-1}$ P$_2$O$_5$ were applied to the field. When coriander plants reached the stem elongation period, second dose of fertilizer i.e., 25 kg N ha$^{-1}$ was applied by using urea as a source. Each plot had four rows and plot length was measured as 4 m. In both the experimental fields, weed management was done according to traditional agricultural practices when needed. Fields were irrigated with drip irrigation according to the crop requirement. After flowering period, the plants were not irrigated.

Split plot design was used, the main plots consisted of coriander varieties (Erbaa and Gamze) and sub plots consisted of seaweed fertilizer doses. Seaweed fertilizer (Entex-A liquid) (Deva Agro Inc. Turkey) was applied in two splits: 40 days after the sowing and at the 60$^{th}$ day (starting of flowering period). The doses were applied as foliar spray containing 0.5 ml L$^{-1}$, 1 ml L$^{-1}$, 2 ml L$^{-1}$ liquid extracts from seaweed leaves and a the control plots were maintained without any treatment. The seaweed applied fertilizer was in liquid form and had 10% organic compounds, 0.4% alginic acid, 9.69 ds/m EC, and a pH of 7.8–9.8. Coriander plants were harvested when the seeds turned brown on 04 September 2021 and plant height (cm), crude protein content (%), seed yield (kg ha$^{-1}$), essential oil yields (L ha$^{-1}$), and essential oil ratios (%) were determined. The two middle lines in every plot were harvested and used for determining the final yield. Essential oil yields (L ha$^{-1}$) were calculated with multiplication of detected percentage of essential oil by obtained seed yields.

**Table 2. Climatic data for the research sites (1929–2021).**

| Months | Average temperature (˚C) | | Total precipitation (mm) | |
|---|---|---|---|---|
| | 2021 | 1929–2020 | 2021 | 1929–2020 |
| May | 22.0 | 18.0 | 20.6 | 45.6 |
| June | 24.7 | 23.1 | 9.1 | 17.4 |
| July | 29.2 | 27.0 | 0.2 | 3.9 |
| August | 28.2 | 27.0 | 23.2 | 3.5 |
| September | 22.8 | 22.5 | 5.7 | 8.1 |
| **Average** | 25.4 | 23.5 | -- | -- |
| **Total** | -- | -- | 58.8 | 78.5 |

## Protein content (%)

Protein content (%) in each plot was calculated according to the Dumas method [29]. For this method, coriander samples were digested at 1200°C in an oven with oxygen gas and the digested samples were oxidized with organic elements. Then, the combusted gasses ($O_2$, $CO_2$, $H_2O$, $N_2$ and nitrogen oxides $NO_x$) were kept together and passed through traps. All the gases were eliminated excluding nitrogen and nitrogen oxide. Then, the gases were transformed into bonded nitrogen molecules or nitrogen oxides and they were carried into an oxidizer catalytic oven with a vector gas. Purification of combustion gases and dehydration: tungsten or copper compounds were applied to these gases for reduction reaction and all the nitrogen compounds were transformed into $N_2$ forms. Detection was done by Thermal Conductivity Detector (TCD). A connected computer calculated the nitrogen concentration from the sample weight by using the signals of TCD. Crude protein content was calculated by multiplication of measured nitrogen amount with factor (6.25) and presented in percentage.

## Extraction of essential oils

The essential oil was extracted using a 1 L Clevenger distillation apparatus. For this purpose, 50 g dried aerial parts were diluted with 500 ml distilled water (1:10 w/v) for 3 h from each treatment. Essential oil was dried over anhydrous sodium sulphate and stored in dark glass bottle at 4°C until used for further analyses [30].

The components of the essential oils of the plants were determined by gas-chromatographic method. Determination of essential oil components were carried out with Thermo Scientific ISQ Single Quadrupole model gas chromatographic device under the following conditions. TR-FAME MS model, 5% Phenyl Polysilphenylene siloxane, 0.25 mm inner diameter × 60 m length, 0.25 μm film thickness column was used. Helium (99.9%) was used as the carrier gas at a flow rate of 1 mL min$^{-1}$. The ionization 22 energy was set at 70 eV, and the mass range m/z was 1.2–1200 amu. Scan Mode was used for data collection. The MS transfer line temperature was 250°C, the MS ionization temperature was 220°C, and the injection port temperature was 220°C. The initial temperature of the column was 50°C which was increased to 220°C with a rate of 3°Cmin$^{-1}$. The structure of each compound was identified using mass spectra with the Xcalibur program (Wiley 9) [31].

## Statistical analysis

The data were analyzed statistically using the General Linear Model one way analysis of variance (ANOVA). Significance of the results was determined according to p values ($p < 0.05$ = significant, $p < 0.01$ = moderate significant, $p < 0.001$ = highly significant). The SPSS 25.0 package program was used in the analysis of variance.

## Results and discussion

Seaweed fertilizer applied on the plants at 40$^{th}$ and 60$^{th}$ days after sowing with different doses showed a significant increase in plant height, essential oil content (%), essential oil yield (L ha$^{-1}$) and protein content (%) of two coriander varieties at different locations.

When the effects of the seaweed fertilizer were considered, there were some significant differences in yield ($p<0.001$), plant height, protein content (%) ($p<0.01$) and essential oil percentage ($p< 0.05$) between the plants in different locations. However, there was nonsignificant difference regarding yield of essential oils (Table 3).

When the coriander plants were compared in location A, which had more clay and organic matter with a low sand percentage, with the plants in location B, which had less organic matter

**Table 3. Effects of seaweed applied at different times and doses at two different locations (± standard error).**

| | Yield (kg ha$^{-1}$) | Plant height (cm) | Essential oil (%) | Essential oil yield (L ha$^{-1}$) | Protein content (%) |
|---|---|---|---|---|---|
| **Locations** | | | | | |
| Location A | 1407.08±25.11a | 78.40±0.96a | 0.45±0.02a | 6.36±0.36 | 20.59±0.22a |
| Location B | 1137.88 ±16.17b | 65.70 ±0.53b | 0.66±0.04b | 7.56±0.45 | 19.03±0.25b |
| F ratio | 524.59*** | 43.09** | 19.57* | 4.163$^{ns}$ | 82.86** |
| **Variety** | | | | | |
| Gamze | 1343.13 ±30.17a | 69.70 ±0.81a | 0.53±0.03a | 7.00 ±0.431 | 20.18±0.27a |
| Erbaa | 1201.83±23.25b | 74.40±1.48b | 0.58 ± 0.03b | 6.90±0.396 | 19.44±0.24b |
| F ratio | 146.00*** | 7.85* | 9.143* | 0.30$^{ns}$ | 28.17** |
| **Application time** | | | | | |
| 40. days | 1312.50± 33.00a | 72.80 ±1.13 | 0.53±0.03 | 6.93±0.41 | 19.73±0.27 |
| 60. days | 1232.46 ± 22.49b | 71.30 ±1.28 | 0.57±0.04 | 6.99± 0.42 | 19.88±0.25 |
| F ratio | 31.34** | 5.56$^{ns}$ | 2.34$^{ns}$ | 0.64$^{ns}$ | 1.87$^{ns}$ |
| **Seaweed doses (ml L$^{-1}$)** | | | | | |
| 2 | 1387.08±38.00a | 73.5±1.68a | 0.65± 0.05a | 8.82±0.60a | 21.19± 0.26a |
| 1 | 1326.25±38.90ab | 72.9± 1.81a | 0.67±0.04a | 8.72±0.43a | 20.40±0.23b |
| 0.5 | 1247.42±37.14b | 73.0±1.81a | 0.51±0.04b | 6.21±0.33b | 19.79±0.23b |
| Control | 1129.17±27.68c | 68.8±1.43b | 0.37±0.03c | 4.09±0.27c | 17.85± 0.34c |
| F ratio | 112.83*** | 12.22*** | 25.24*** | 33.98*** | 42.68*** |

*$p < 0.05$ (significant)

**$p < 0.01$ (moderate significant)

***$p < 0.001$ (highly significant); ns, non-significant. The difference between the averages represented with the same letter in the same column and same group was statistically insignificant.

and clay with high sand percentage, it was recorded that in location A plants the yield was increased by 20%, the plant height by 16% and the protein content by 7.5%. However, for location B plants, there was opposite situation for essential oil percentage and essential oil yield (about 72 and 16%, respectively) (Table 3). The response of Gamze and Erbaa varieties to the applied seaweed fertilizer was significant considering the plant height, yield (p<0.001) and protein content (p<0.01) while the difference in essential oil yield was not significant (Table 3).

Seaweed application at 40[th] and 60[th] days after the sowing showed significant effect on yield (p<0.01). The seaweed application caused an increase in yield, essential oil content and protein content (%) for Gamze variety but for Erbaa variety, only plant height and essential oil content were increased. Although, there were no effects on the plant height, essential oil yield and protein content. Total 8% increase in yield of coriander was obtained when the seaweed had been applied at the 40[th] and 60[th] days after sowing. It was claimed that when the seaweed was applied at different times and with different dosages, it had really important effects for yield, plant height, essential oil percentage, essential oil yield and protein percentages (p< 0.001). The dose 2 ml L$^{-1}$ caused an increase in all parameters as compared to the control treatments (without seaweed application). It was noted that with the seaweed application, coriander yield was increased by 19%, plant height by 6.5%, essential oil by 43%, essential oil yield by 54% and protein percentage by 16%. The foliar application of seaweed caused positive effects on yield, plant height, essential oil percentage, essential oil yield and protein percentage (Table 3).

Among the results obtained in the experiment, interactions at different levels of significant (p< 0.05, p< 0.01, p< 0.001) were found between the yield, plant height, essential oil rate,

**Table 4. Interactions of seaweed applied at different times and doses and at two different locations (± standard error).**

| | | Yield (kg ha$^{-1}$) | Plant height (cm) | Essential oil (%) | Essential oil yield (L ha$^{-1}$) | Protein content (%) |
|---|---|---|---|---|---|---|
| Varieties × Locations | | | | | | |
| Gamze | Location A | 1503.33±32.98a | 74.3±0.58 | 0.42±0.03 | 6.35±0.50 | 20.98±0.35 |
| | Location B | 1182.92±20.01c | 65.0±0.70 | 0.64± 0.05 | 7.67±0.69 | 19.38±0.35 |
| Erbaa | Location A | 1310.83±26.15b | 82.6±1.40 | 0.48±0.04 | 6.36 ± 0.53 | 20.20±0.24 |
| | Location B | 1092.83 ±22.17d | 66.3±0.78 | 0.68± 0.02 | 7.45± 0.58 | 18.68±0.35 |
| | *F ratio* | 19.31* | 4.19$^{ns}$ | 0.57$^{ns}$ | 0.34$^{ns}$ | 0.07$^{ns}$ |
| Application time × Locations | | | | | | |
| 40. days | Location A | 1472.92±39.25a | 78.80 ± 1.21 | 0.45±0.03 | 6.77±0.55ab | 20.49±0.30 |
| | Location B | 1152.08±25.95c | 66.80 ±0.81 | 0.61±0.05 | 7.08±0.63ab | 18.98±0.38 |
| 60. days | Location A | 1341.25±25.69b | 78.10 ±1.53 | 0.44± 0.03 | 5.94±0.46b | 20.69±0.32 |
| | Location B | 1123.67±109.44c | 64.5± 0.61 | 0.70± 0.05 | 8.04±0.63a | 19.08±0.33 |
| | *F ratio* | 13.04** | 1.82$^{ns}$ | 3.95$^{ns}$ | 6.04* | 0.25$^{ns}$ |
| Locations × Seaweed doses | | | | | | |
| Location A | 2 | 1521.67±50.82a | 80.4±1.46 | 0.52± 0.05bcd | 7.85±0.77 | 22.00± 0.37a |
| | 1 | 1470.83±46.82ab | 79.4±2.03 | 0.53±0.04bc | 7.66±0.61 | 20.75±0.31b |
| | 0.5 | 1399.17±34.52b | 79.6±2.23 | 0.40± 0.03de | 5.60± 0.46 | 20.45±0.24b |
| | Control | 1236.67± 25.62c | 74.3±1.59 | 0.35± 0.04e | 4.31±0.49 | 19.15±0.37c |
| Location B | 2 | 1252.50±21.99c | 66.6±0.97 | 0.78± 0.07a | 9.78±0.88 | 20.38±0.16b |
| | 1 | 1181.67±19.77cd | 66.3±1.35 | 0.83± 0.05a | 9.78±0.88 | 20.05±0.30b |
| | 0.5 | 1095.67±19.73de | 66.5± 0.97 | 0.63± 0.05b | 6.32±0.42 | 19.13±0.30c |
| | Control | 1021.67± 21.14e | 63.2±0.53 | 0.38± 0.03de | 3.88±0.25 | 16.55±0.20d |
| | *F ratio* | 3.42** | 0.81$^{ns}$ | 4.53** | 2.22$^{ns}$ | 3.3* |
| Application time × Seaweed doses | | | | | | |
| 40.days | 2 | 1468.33±60.19a | 73.1± 2.00 | 0.58±0.06 | 8.53± 0.85 | 21.28± 0.34 |
| | 1 | 1360.83±65.45ab | 75.1±2.32 | 0.70 ± 0.07 | 9.08 ± 0.65 | 20.35± 0.33 |
| | 0.5 | 1291.67±59.52bc | 74.2±2.46 | 0.48 ± 0.04 | 6.00± 0.36 | 19.45±0.40 |
| | Control | 1129.17± 40.03d | 68.8 ± 2.06 | 0.37 ± 0.04 | 4.09± 0.39 | 17.85 ± 0.49 |
| 60. days | 2 | 1305.83± 39.21bc | 73.9±2.79 | 0.72 ± 0.08 | 9.10±0.88 | 21.10±0.40 |
| | 1 | 1291.67±42.74bc | 70.7 ± 2.73 | 0.66±0.05 | 8.36±0.58 | 20.45±0.32 |
| | 0.5 | 1203.17±42.67cd | 71.9±2.72 | 0.55 ± 0.06 | 4.09± 0.39 | 20.13 ± 0.22 |
| | Control | 1129.167± 40.03d | 68.8±2.06 | 0.37 ± 0.04 | 4.09± 0.39 | 17.85 ± 0.49 |
| | *F ratio* | 10.16*** | 3.46$^{ns}$ | 1.86$^{ns}$ | 0.56$^{ns}$ | 0.71$^{ns}$ |

*$p < 0.05$ (significant)

**$p < 0.01$ (moderate significant)

***$p < 0.001$ (highly significant); ns, non-significant. The difference between the averages represented with the same letter in the same column and same group was statistically insignificant.

essential oil yield and protein content of coriander. In the Locations x varieties interaction, Gamze and Erbaa coriander cultivars yielded higher yields in Location A, which has high clay and organic matter content and low sand content, while higher yield was obtained from Gamze variety in this interaction (Table 4). Locations × time interaction: it was obtained that when the seaweed was applied at the 40[th] day after sowing, the yield in location A (high clay ratio and high amount of organic matter) had reached the highest value. The second highest yield was obtained from the seaweed application at the 60[th] day after sowing at location A. At location B, the yield values were lower when they were compared to location A plants at the fertilizer application times (40[th] and 60[th] days after sowing). The highest essential oil yield was

obtained from the seaweed application at the 60th day after sowing at location B and, the second highest essential oil yield was obtained from the seaweed application at the 40th day in location B (8.04 ± 0.63, 7.08 ± 0.63, respectively) (Table 4).

In case of location × seaweed doses interaction, the application of seaweed fertilizer at different times and with different doses caused an increase in yield, plant height and protein content in location A as compared to location B. However, it was not the same for essential oil yield and essential oil ratio. According to soil type, it was found that the essential oil yield and contents were higher at location B. Additionally, in both the locations (location A and location B) when the doses of seaweed were increased, it caused increase in the measured values with positive effects. The highest yield was obtained from 2 ml L$^{-1}$ seaweed application in both locations (except essential oil ratio) (Table 4).

Time of application × seaweed doses interactions: when the seaweed was applied, it caused significant differences in yield. The highest yield belonged to 2 ml L$^{-1}$ seaweed dose at the 40th day after sowing; and 1 ml L$^{-1}$ application of seaweed had the second place for yield. When the seaweed is applied on coriander, it causes increase in yield independently from location and variety.

There was a positive interaction between application time × location × seaweed doses and the highest yield was obtained from the seaweed applications with different doses at the 40th day after the sowing at location A. At the 60th day after sowing, seaweed application with different doses also caused increase in yield at location A. In both the locations and at both the application times, the highest yield had been obtained from 2 ml L$^{-1}$ seaweed application. Even though, the seaweed application at the 40th and at the 60th days did not increase the yield much in location B. But it caused increase in essential oil ratio and essential oil yield. In case of application of 1 ml L$^{-1}$ seaweed after 40 days from sowing and 2 ml L$^{-1}$ seaweed application 60 days after the sowing caused a significant increase in yield and ratio of essential oil. In location B (with low amount of organic matter and high sand ratio) the yield was lower than location A but, the yield and ratio of essential oil were higher (Table 5).

According to locations with different soil structures, the positive effects of seaweed applications on coriander plants were found. Sand, clay and organic matter contents in the soil significantly affected the essential oil ratio and yield regardless of the variety of coriander plant. The soil which was sandy, and had low amount of clay and organic matter, increased the yield of essential oil and ratio of essential oil. Yet, the other soil type (rich in organic matter and clay, poor in sand) caused decrease in essential oil yield and ratio. Yield of coriander plant may depend on climate, genetics, ecological conditions and variety [32,33]. While, on the other hand, plant height can be affected by ecological conditions and crop management methods alongside the genotype features [28]. In some studies, yield of coriander changed between 532–2713 kg ha$^{-1}$ [28,34,35] and Ozel et al. [36] claimed that plant height was between 34.50–111.63 cm. Moreover, essential oil ratios may vary between 0.13–1.6% [28,37–39] and protein content may range between 14.0–19.34% [40,41]. The values of yield, plant height, yield and content of essential oil that were studied in the literature were in the same range with our studies, while the protein ratios that we obtained were higher when compared with these studies. Besides, there were some differences in plant height, essential oil yield and protein percentages between the coriander varieties but these differences might be caused by genotype, soil and climate conditions, chemical fertilizer, and species type [28].

The high sand content in the soil caused an increase in the essential oil ratio and yield (location B) and these results are similar to other studies, which showed that soil stress conditions like high pH, drought, low nutrients, sand, and salinity caused an increase in essential oil ratios [42–44]. Likewise, Aboukhalid et al. [45] reported that the quality of essential oils was affected by soil structure. Aziz et al. [46] and Said-Al Ahl et al. [47] also observed that essential oil

**Table 5. Application time × locations × seaweed doses interactions of seaweed applied at different times and doses in two different locations (± standard error).**

| Application time × Locations × Seaweed doses | | | Yield (kg ha⁻¹) | Plant height (cm) | Essential oil (%) | Essential oil yield (L ha⁻¹) | Protein content (%) |
|---|---|---|---|---|---|---|---|
| 40. days | Location A | 2 | 1635.00 ± 64.49a | 79.20±0.70 | 0.55±0.08cde | 8.68±1.192bcd | 21.95±0.53 |
| | | 1 | 1545.00 ± 63.18ab | 80.80±2.68 | 0.52 ± 0.06cde | 7.86±0.799cde | 20.60±0.39 |
| | | 0.5 | 1475.00 ± 35.94bc | 80.70±2.82 | 0.40±0.04e | 5.93±0.614efgh | 20.25± 0.40 |
| | | Control | 1236.67± 38.01ef | 74.30±2.36 | 0.35±0.06e | 4.31±0.719gh | 19.15±0.55 |
| | Location B | 2 | 1301.67±25.87def | 67.00±1.51 | 0.62±0.10bcd | 8.08±1.307bcde | 20.60±0.25 |
| | | 1 | 1176.67±35.93fg | 69.30±1.82 | 0.88±0.08a | 10.30±0.787ab | 20.10±0.54 |
| | | 0.5 | 1108.33±32.60gh | 67.70±1.28 | 0.55±0.04cde | 6.08±0.451efgh | 18.65±0.53 |
| | | Control | 1021.67±31.35h | 63.20±0.79 | 0.38±0.040e | 3.88±0.367h | 16.55±029 |
| 60. days | Location A | 2 | 1408.33 ±45.49cd | 81.70±2.88 | 0.48±0.07de | 6.73±0.818defg | 22.05±0.57 |
| | | 1 | 1396.67 ±56.67cd | 78.00 ± 3.18 | 0.53±0.06cde | 7.46±0.990cdef | 20.90±0.52 |
| | | 0.5 | 1323.33±40.63de | 78.50±3.66 | 0.40±0.06d | 5.27±0.717fgh | 20.65± 0.30 |
| | | Control | 1236.67±38.01ef | 74.30± 2.36 | 0.35±0.06e | 4.31±0.719gh | 19.15±0.55 |
| | Location B | 2 | 1203.33± 22.16efg | 66.20± 1.35 | 0.95±0.04a | 11.47±0.715ab | 20.15±0.17 |
| | | 1 | 1186.67±20.44fg | 63.30±1.02 | 0.78±0.05ab | 9.25±0.448abc | 20.00± 0.33 |
| | | 0.5 | 1083.00±24.19gh | 65.30±1.38 | 0.70±0.06bc | 7.56±0.580cdef | 19.60± 0.15 |
| | | Control | 1021.67± 31.35h | 63.20±0.79 | 0.38± 0.04e | 3.88±0.367h | 16.55± 0.29 |
| | F ratio | | 2.817* | 0.52ns | 3.82* | 3.34* | 0.32ns |

*$p < 0.05$ (significant); ns, non-significant. The difference between the averages represented with the same letter in the same column and same group was statistically insignificant.

components ratio of *Thymus vulgaris* L. were higher when cultivated in loamy soil than in clayey soil. Khalid and Ahmed [44] had also similar results on citrus plants. In this study, it was also claimed that when the amount of sand increased, the essential oil ratio and yield also increased.

Seaweed extracts can be applied as foliar spray or directly to the soil. The increase in plant yield by seaweed fertilizer application is related with hormonal components especially cytokinins that are present in seaweeds [48]. On the other hand, in different studies, it has been reported that when the seaweed extract was applied as foliar spray, the application affected root growth positively and as a result of this; plants could get more water and nutrients from the soil; this resulted in the increased yield [49,50]. In this study, plant height, protein content and essential oil ratio, yield and essential oil yield increased because of seaweed application. The increase in coriander plant growth might be related with micro and macro nutrients, cytokinins, auxins and betaines present in seaweed extracts that speed up the photosynthetic rate and help in vegetative growth [51]. In previous studies, it was reported that seaweed caused an increase in yields of other plants. For example, Crouch and Van Staden [52] claimed that with seaweed application, yield of tomato plants increased by 30% and, Basavaraja et al. [53] stated that they got 26% increase in yield of hybrid corns by seaweed application. Also, in this study, when the seaweed had been applied under the suitable conditions (in location A at the 40th day, 2 ml L⁻¹) the yield increased by 25% (Table 5). Similarly, with seaweed applications, the yields were also increased in some plants like soybean (*Glycine max* (L.) Merr.) [54], strawberry (*Fragaria ananassa* (Duchesne ex Weston) Duchesne ex Rozier) [55] or watermelon (*Citrullus lanatus* (Thunb.) Matsum. & Nakai) [56]. Application of seaweed increased essential oil ratios and similar results were also claimed by Jhariya and Jain [57], for coriander, Garg [58], for fennel and Gharib et al. [59] for marjoram; using organic and biofertilizers increased the ratio of essential oil. Moreover, in this study it was found that by seaweed application,

increase in protein of coriander plants was achieved. In previous studies, it was determined that coriander protein percentages ranged between 11.3%-21.3% [40,60] and in this study the results are similar to previous studies. Especially, the seaweed caused a significant increase in protein percentages when the plants were compared with the control treatment.

## Essential oils contents

Data for essential oil contents showed that the coriander plants cultivated at two different locations had different amounts of essential oil compounds. In location A plants, there were 22 components (Table 6) and in location B plants, there were 21 components (Table 7). Linalool, α-pinene, geranyl aacetate and γ-terpinene were determined as the major components. Linalool, an important component of coriander, was obtained in higher concentration from location B which had more of sand as a soil component. Among the varieties, more linalool contents were obtained in the Erbaa variety. Depending on the locations, there were some differences in the components; linalool ratios were 77.25–87.29%, α-pinene ratios were 2.03–5.66%, γ-terpinene ratios were 1.75–4.28%, and geranly acetate ratios were 1.94–5.81%. The seaweed application (2 ml L$^{-1}$) at the 40th and 60th days after sowing caused the highest amount of linalool at location A plants (for both Gamze and Erbaa varieties). Still, both the coriander varieties which had been applied with seaweed had higher linalool when compared with the

**Table 6. Essential oil ratios of the two coriander varieties applied with different seaweed doses at different times (day 40 and day 60) at location A.**

| | Varieties | | | Gamze | | | | | | | Erbaa | | | | | | |
| | Application time | | | 40.days | | | 60. days | | | | 40. days | | | 60. days | | | |
| | Seaweed doses (ml L$^{-1}$) | | | 2 | 1 | 0.5 | 2 | 1 | 0.5 | Control | 2 | 1 | 0.5 | 2 | 1 | 0.5 | Control |
| RT | Compound Name | SI | RSI | | | | | | | | | | | | | | |
| *6.57* | *α-Pinene* | *990* | *994* | *2.04* | *5.66* | *3.98* | *3.28* | *3.92* | *2.09* | *3.83* | *4.37* | *2.60* | *4.20* | *3.85* | *2.55* | *4.19* | *3.28* |
| 7.58 | Camphene | 988 | 991 | - | - | 0.24 | - | - | - | 0.21 | - | - | - | 0.33 | 0.12 | 0.20 | - |
| 7.57 | β-Pinene | 776 | 788 | 0.49 | 0.42 | 0.31 | 0.31 | 0.28 | 0.14 | 0.28 | 0.28 | 0.20 | 0.29 | 0.45 | 0.21 | 0.30 | 0.25 |
| 8.69 | β-phellandrene | 974 | 986 | 0.10 | 0.13 | 0.12 | 0.10 | 0.11 | 0.06 | 0.11 | 0.11 | 0.08 | 0.11 | 0.16 | 0.08 | 0.11 | 0.09 |
| 8.99 | β-myrcene | 986 | 994 | 0.29 | 0.33 | 0.37 | 0.29 | 0.32 | 0.20 | 0.32 | 0.26 | 0.24 | 0.31 | 0.36 | 0.25 | 0.28 | 0.22 |
| 10.03 | Limonene | 990 | 994 | 0.68 | 0.78 | 0.81 | 0.65 | 0.70 | 0.48 | 0.71 | 0.64 | 0.56 | 0.70 | 0.66 | 0.53 | 0.64 | 0.56 |
| *11.10* | *γ-terpinene* | *995* | *995* | *1.75* | *4.24* | *4.28* | *2.65* | *3.97* | *2.79* | *3.83* | *3.23* | *3.18* | *3.78* | *3.46* | *3.00* | *3.68* | *3.25* |
| 11.98 | Terpinolene | 915 | 931 | - | 0.25 | 0.24 | 0.21 | 0.24 | 0.16 | 0.20 | 0.18 | 0.17 | 0.21 | 0.23 | 0.15 | 0.18 | 0.16 |
| 12.00 | Eucalyptol | 909 | 919 | 0.69 | - | - | - | - | - | - | - | - | - | - | - | - | - |
| 12.68 | Cymene | 975 | 993 | 0.65 | 1.09 | 1.08 | 0.76 | 0.88 | 0.58 | 0.98 | 0.91 | 0.81 | 1.09 | 0.70 | 0.80 | 1.11 | 0.83 |
| *20.63* | *Linalool* | *997* | *998* | *82.03* | *77.25* | *76.39* | *83.18* | *80.43* | *80.18* | *78.01* | *82.61* | *84.98* | *80.80* | *83.35* | *82.93* | *80.70* | *80.74* |
| 22.81 | Decanal | 984 | 993 | 0.13 | 0.23 | 0.26 | 0.21 | 0.19 | 0.13 | 0.17 | 0.12 | 0.14 | 0.22 | 0.24 | 0.37 | 0.20 | 0.27 |
| 24.26 | Terpinen-4-ol | 977 | 985 | 0.16 | 0.07 | 0.14 | 0.09 | 0.10 | 0.12 | 0.13 | 0.08 | 0.13 | 0.09 | 0.09 | 0.16 | 0.11 | 0.11 |
| 25.94 | Camphor | 984 | 990 | 1.71 | 1.39 | 2.01 | 1.64 | 1.64 | 1.72 | 1.95 | 1.52 | 1.81 | 1.51 | 1.75 | | 1.87 | 1.73 |
| 27.04 | α-Terpineol | 900 | 981 | 0.41 | 0.10 | 0.23 | 0.17 | 0.17 | 0.18 | 0.18 | 0.12 | 0.17 | 0.13 | 0.11 | 0.23 | 0.16 | 0.14 |
| 27.51 | Borneol | 969 | 984 | 0.31 | - | 0.11 | - | - | 0.09 | 0.10 | - | 0.09 | 0.07 | - | - | - | 0.09 |
| 27.56 | Myrtenylacetate | 726 | 754 | - | 0.06 | - | 0.06 | - | - | - | 0.06 | - | - | 0.10 | - | - | - |
| *28.54* | *GeranylAcetate* | *988* | *988* | *1.94* | *4.38* | *4.15* | *2.90* | *3.64* | *2.49* | *3.56* | *3.68* | *3.04* | *3.60* | *2.54* | *4.35* | *4.32* | *3.45* |
| 28.77 | trans-2-Decenal | 904 | 949 | 0.03 | 0.07 | 0.14 | 0.06 | 0.04 | 0.02 | 0.03 | 0.09 | 0.10 | 0.17 | 0.15 | 0.22 | 0.09 | 0.23 |
| 30.36 | Geranıol | 980 | 983 | 0.62 | 0.25 | 0.63 | 0.57 | 0.57 | 0.58 | 0.67 | 0.54 | 0.71 | 0.40 | 0.36 | 0.91 | 0.76 | 0.66 |
| 35.22 | 2-Dodecenal | 959 | 975 | 0.14 | 0.22 | 0.32 | 0.18 | 0.18 | 0.09 | 0.14 | 0.20 | 0.17 | 0.23 | 0.37 | 0.26 | 0.22 | 0.27 |
| 36.80 | Methylcinnamate | 990 | 992 | 3.88 | 1.13 | 3.37 | 1.80 | 1.59 | 2.25 | 0.02 | 0.10 | 0.17 | 1.19 | 0.02 | 0.07 | 0.19 | 0.02 |
| | Others | | | 1.6 | 0.73 | 0.72 | 0.55 | 0.58 | 0.41 | 0.38 | 0.47 | 0.34 | 0.75 | 7.39 | 1.25 | 0.03 | 0.18 |
| | Total | | | 99.65 | 98.8 | 99.9 | 99.66 | 99.55 | 94.76 | 95.81 | 99.57 | 99.74 | 99.85 | 99.67 | 98.44 | 99.34 | 96.53 |

**Table 7. Essential oil ratios of the two coriander varieties applied with different seaweed doses at different times (day 40 and day 60) at location B.**

| | Varieties | | | Gamze | | | | | | | Erbaa | | | | | | |
| --- | --- | --- | --- | --- | --- | --- | --- | --- | --- | --- | --- | --- | --- | --- | --- | --- | --- |
| | Application time | | | 40. days | | | 60. days | | | | 40. days | | | 60. days | | | |
| | Seaweed doses (ml L$^{-1}$) | | | 2 | 1 | 0.5 | 2 | 1 | 0.5 | Control | 2 | 1 | 0.5 | 2 | 1 | 0.5 | Control |
| RT | Compound Name | SI | RSI | | | | | | | | | | | | | | |
| *6.57* | *α-Pinene* | *993* | *995* | *5.13* | *4.84* | *4.68* | *3.17* | *3.33* | *2.65* | *3.27* | *2.03* | *2.48* | *5.40* | *2.15* | *2.70* | *3.17* | *2.36* |
| 7.56 | Camphene | 981 | 986 | 0.25 | 0.28 | 0.27 | - | - | - | - | 0.19 | - | 0.30 | - | - | - | - |
| 8.39 | β-Pinene | 971 | 985 | 0.37 | 0.35 | 0.38 | 0.43 | 0.29 | 0.21 | 0.23 | 0.28 | 0.19 | 0.38 | 0.17 | 0.20 | 0.21 | 0.16 |
| 8.69 | β-phellandrene | 972 | 975 | 0.13 | 0.12 | 0.11 | 0.09 | 0.09 | 0.07 | 0.08 | 0.11 | 0.08 | 0.14 | 0.07 | 0.08 | 0.09 | 0.08 |
| 8.99 | β-myrcene | 982 | 994 | 0.35 | 0.33 | 0.35 | 0.28 | 0.26 | 0.22 | 0.25 | 0.29 | 0.20 | 0.40 | 0.16 | 0.21 | 0.23 | 0.17 |
| 10.02 | Limonene | 988 | 995 | 0.88 | 0.87 | 0.87 | 0.64 | 0.61 | 0.52 | 0.66 | 0.69 | 0.51 | 0.95 | 0.40 | 0.55 | 0.58 | 0.42 |
| *11.09* | *γ-terpinene* | *994* | *995* | *3.22* | *3.66* | *3.72* | *3.07* | *2.54* | *2.37* | *2.57* | *2.53* | *2.91* | *4.28* | *2.29* | *2.93* | *3.41* | *2.67* |
| 11.95 | Terpinolene | 941 | 976 | 0.19 | 0.21 | 0.22 | 0.17 | 0.18 | 0.14 | 0.16 | 0.18 | 0.15 | 0.22 | 0.12 | 0.15 | 0.16 | 0.12 |
| 12.65 | Cymene | 985 | 991 | 1.03 | 1.91 | 2.05 | 0.88 | 0.78 | 0.72 | 1.56 | 0.33 | 0.77 | 1.79 | 0.75 | 1.02 | 0.96 | 0.76 |
| 18.76 | Linalooloxide | 879 | 924 | 0.12 | 0.03 | 0.07 | 0.05 | 0.04 | 0.02 | 0.09 | 0.05 | 0.02 | 0.09 | 0.04 | 0.05 | 0.06 | 0.02 |
| *20.58* | *Linalool* | *996* | *997* | *80.15* | *79.52* | *77.26* | *83.48* | *84.31* | *81.73* | *78.73* | *87.15* | *85.88* | *78.81* | *87.29* | *85.25* | *83.72* | *81.49* |
| 22.78 | Decanal | 991 | 997 | 0.35 | 0.30 | 0.38 | 0.28 | 0.34 | 0.26 | 0.34 | 0.26 | 0.18 | 0.28 | 0.22 | 0.21 | 0.24 | 0.17 |
| 24.25 | Terpinen-4-ol | 959 | 965 | 0.09 | 0.08 | 0.08 | 0.12 | 0.13 | 0.09 | 0.09 | 0.11 | 0.11 | 0.10 | 0.09 | 0.11 | 0.11 | 0.09 |
| 25.90 | Camphor | 984 | 989 | 1.60 | 1.42 | 1.61 | 1.66 | 1.49 | 1.47 | 1.36 | 1.71 | 1.65 | 1.64 | 1.49 | 1.71 | 1.63 | 1.43 |
| 27.06 | α-Terpineol | 963 | 980 | 0.13 | 0.08 | 0.10 | 0.17 | 0.16 | 0.15 | 0.12 | 0.12 | 0.15 | 0.12 | 0.14 | 0.16 | 0.15 | 0.13 |
| 27.52 | Borneol | 831 | 875 | 0.09 | 0.09 | - | 0.09 | 0.11 | 0.09 | 0.10 | 0.09 | 0.06 | 0.09 | 0.07 | 0.09 | 0.06 | 0.04 |
| *27.72* | *Geranylacetate* | *869* | *924* | *3.42* | *4.27* | *5.81* | *3.64* | *3.57* | *3.53* | *3.89* | *2.25* | *2.86* | *3.43* | *2.75* | *2.83* | *3.37* | *2.40* |
| 27.96 | 1-Decanol | 891 | 918 | 0.13 | 0.08 | 0.12 | - | 0.13 | - | 0.12 | 0.07 | - | 0.10 | 0.07 | - | - | - |
| 28.77 | trans-2-Decenal | 904 | 949 | 0.18 | 0.23 | 0.29 | 0.10 | 0.14 | 0.17 | 0.24 | 0.12 | 0.09 | 0.17 | 0.11 | 0.13 | 0.12 | 0.10 |
| 30.36 | Geraniol | 980 | 983 | 0.43 | 0.29 | 0.41 | 0.68 | 0.65 | 0.62 | 0.41 | 0.55 | 0.71 | 0.43 | 0.68 | 0.64 | 0.71 | 0.61 |
| 35.22 | 2-Dodecenal | 959 | 975 | 0.48 | 0.43 | 0.46 | 0.28 | 0.33 | 0.42 | 0.35 | 0.23 | 0.26 | 0.31 | 0.27 | 0.29 | 0.36 | 0.27 |
| | Others | | | 0.53 | 0.46 | 0.62 | 0.56 | 0.46 | 0.86 | 0.60 | 5.41 | 0.38 | 0.43 | 0.41 | 0.59 | 0.40 | 1.29 |
| | Total | | | 99.25 | 99.85 | 99.86 | 99.84 | 99.94 | 96.01 | 95.22 | 99.75 | 99.64 | 99.86 | 99.74 | 99.90 | 99.74 | 94.78 |

control treatment (Table 6). At location B, the highest linalool amount was obtained from Gamze variety with 1 ml L$^{-1}$ seaweed dose at the 60th day after sowing but for Erbaa variety, the highest amount of linalool was obtained from 2 ml L$^{-1}$ seaweed application at the 40th and at the 60th days after sowing (Table 7). In different studies on coriander plants, linalool ratio ranges were reported as; Delaquis et al. [61] 69.80%, Ozel et al. [36] 76.12–82.74%, Gucuk [62], 53–74%, and Beyzi and Gunes [63], by 83.81–91.40%. The concentration of linalool ratio was supported by Izgi [35] who stated that the changes in linalool ratios might be related with ecological, traditional and variety differences. In this study, the higher linalool ratio (compared to control) is related with seaweed application. Ozyazici, [28] reported that organo-mineral and organic fertilizers increased linalool ratio.

## Conclusion

This study analyzed the effects of seaweed foliar fertilizer on yield, protein content, essential oil ratio, and essential oil yield of coriander plants (two different varieties) cultivated at two different locations. Results showed that the concentration of coriander essential oil was higher in sandy soil and seaweed applications had significant effects on yield, yield components and essential oil components. Among the varieties, more yields were obtained from Gamze variety. The highest yield was obtained from 2 ml L$^{-1}$ seaweed application at the 40th day after sowing in location A which had more clay and organic matter in the soil. The highest essential oil

concentration and yield was obtained from 2 ml L$^{-1}$ seaweed application at the 60$^{th}$ day after sowing at location B which had sandy soil. This study demonstrates that foliar application of seaweed as an organic fertilizer has significant impacts in terms of increasing the production of coriander.

## Author Contributions

**Conceptualization:** Ayse Ozlem Tursun.

**Data curation:** Ayse Ozlem Tursun.

**Formal analysis:** Ayse Ozlem Tursun.

**Investigation:** Ayse Ozlem Tursun.

**Methodology:** Ayse Ozlem Tursun.

**Validation:** Ayse Ozlem Tursun.

**Writing – original draft:** Ayse Ozlem Tursun.

**Writing – review & editing:** Ayse Ozlem Tursun.

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
