## [Decision Letter · Decision Letter 0]

19 Apr 2022

PONE-D-22-06119Effect of seaweed organic fertilizer as foliar spray on yield, essential oil and chemical composition of corianderPLOS ONE

Dear Dr. TURSUN,

Thank you for submitting your manuscript to PLOS ONE. After careful consideration, we feel that it has merit but does not fully meet PLOS ONE’s publication criteria as it currently stands. Therefore, we invite you to submit a revised version of the manuscript that addresses the points raised during the review process.

We look forward to receiving your revised manuscript.

Kind regards,

Sajid Ali

Academic Editor

PLOS ONE

Journal Requirements:

Reviewers' comments:

Reviewer's Responses to Questions

**Comments to the Author**

1. Is the manuscript technically sound, and do the data support the conclusions?

Reviewer #1: No

Reviewer #2: Partly

2. Has the statistical analysis been performed appropriately and rigorously? 

Reviewer #1: Yes

Reviewer #2: Yes

3. Have the authors made all data underlying the findings in their manuscript fully available?

Reviewer #1: No

Reviewer #2: Yes

4. Is the manuscript presented in an intelligible fashion and written in standard English?

Reviewer #1: No

Reviewer #2: No

5. Review Comments to the Author

Reviewer #1: The author proceeds with the use of an "algae-based" fertilizer, which is why it was supposed to have indicated its origin: brand, composition, etc.

The author limited his characterization to this sentence: "The seaweed applied fertilizer which has 10% organic compounds, 0.4% alginic acid"... so it could have any composition, as it does not specify which algae and which production methods are used. behind the production of this "algae-based" fertilizer.

The presence of alginic acid in its composition does not make it a fertilizer based on algae, but based on "alginic acid".

The author will have to redo this part of the manuscript, as it is essential to understand its effect on the production of the aromatic plant.

Some additional corrections are made directly in the manuscript, which is attached to this review.

Reviewer #2: Thank you for the opportunity to review this manuscript. The investigation is useful but the study itself suffers from some conceptual flaws, which are noted in more detail in the attached review

The manuscript needs extensive revision for language and grammar.

As a first step, I advise the authors to find a native English speaker to proofread the manuscript.

Title

The title is good, but you can use a more attention-grabbing title.

Abstract

The abstract should provide a concise and complete summary of the manuscript. ‘‘missed’’.

I am missing the key findings of the study.

I am missing the application and recommendation of the outcomes.

There are important aspects to review regarding some grammar and English usage issues.

The highlights are general, please write more specific

Introduction

Recent references must be cited.

Be careful about citing old references. The rule of thumb is to go back at most five to six years.

Line 45-46 this information isn't important, you can delete it.

Line 48-51 Reference is needed

Line 51-53 please avoid repetition.

Line 58 recent references should be cited

Line 91 should start a new paragraph.

Lines 96-99 Please rewrite this sentence to make it more clear and understandable. ‘‘For this reason, this study was carried out to determine the changes on yield, quality and compounds of essential oils by applying in different doses and at different times (before or initial stage of flowering) of seaweed to two different coriander varieties.’’

In your manuscript, you must include clear research question, aim, and objectives.

What is the significance of your research?

Materials and methods

Methodology section have major limitation, should be more clear

Line 104 ‘tests’ is not appropriate word

Table 2. Main climate data of the research site for the long term (1981–2019) OR 1929-2020

Lines 135-138 confused, please rewrite.

Line 141 Raw Protein Ratio, confused

Line 142 ‘the Dumas method’ reference is needed

Line 155 Extraction of essential oils

Line 155 ‘Reference is needed’

Line 173 The data were under went to statistical analysis?????

The language is ambiguous and confused.

Results and Discussion

Again, I advise the authors to find a native English speaker to proofread the manuscript.

It is better to separate the results and the discussion.

This section can be improved. Please present your key findings in an orderly and logical sequence, without interpretation.

Lines 179-180 ‘‘Seaweed fertilizer was applied on the plants 40th and 60th days after sowing and it was applied with different dosages’’. It should be mentioned in the materials section rather than here.

Lin 324: Nutrients of Volatile Oils????????

Lin 186 ‘‘important’’ is not appropriate word should be ‘‘significant’’.

Discussion

The section is poor. The purpose of the discussion section is to interpret and describe the significance of your findings in relation to what was already known.

Recent references must be included in the manuscript.

https://doi.org/10.1016/j.indcrop.2020.113202

Conclusion

The conclusion should summarize the paper's findings and generalize their importance, discuss ambiguous data, and recommend further studies.

This chapter is more like discussion part. Please revise and strengthen this chapter and provide conclusions and recommendations based on the results of your research.

Some editing for English language is required throughout the manuscript due to too many mistakes such as:

Line 366: ‘‘locationswere’’ change to locations were

Line 371: mora???

Line 377 wafer???

References

Be careful about citing old references. The rule of thumb is to go back at most five to six years.

Best wishes

6. PLOS authors have the option to publish the peer review history of their article (what does this mean?). If published, this will include your full peer review and any attached files.

Reviewer #1: No

Reviewer #2: **Yes: **Mohamed Elsadek

---

## [Author Response · Author response to Decision Letter 0]

9 May 2022

Thank you for arranging a review of the submission. The manuscript is now carefully revised considering all the comments from the two anonymous reviewers and the editor. Please find in following a response to each of the question/suggestion/comment from the reviewers. In addition, references are given by numbering according to the journal style. Thank you for your time and input.

---

## [Editor Report · Decision Letter 1]

16 May 2022

Effect of foliar application of seaweed (organic fertilizer) on yield,essential oil and chemical composition of coriander

PONE-D-22-06119R1

Dear Dr. TURSUN,

We’re pleased to inform you that your manuscript has been judged scientifically suitable for publication and will be formally accepted for publication once it meets all outstanding technical requirements.

Kind regards,

Sajid Ali

Academic Editor

PLOS ONE
---

## [Editor Report · Acceptance letter]

18 May 2022

PONE-D-22-06119R1 

Effect of foliar application of seaweed (organic fertilizer) on yield, essential oil and chemical composition of coriander 

Dear Dr. Tursun:

I'm pleased to inform you that your manuscript has been deemed suitable for publication in PLOS ONE. Congratulations! Your manuscript is now with our production department. 

Kind regards, 

on behalf of

Dr. Sajid Ali 

Academic Editor

PLOS ONE